Reversal of cholestatic liver disease by the inhibition of sphingosine 1-phosphate receptor 2 signaling

Cao Huiling 1
Chen Lin 2
Zeng Ziyang 2
Wu Xianfeng 2
Lei Yuhao 2
Jia Wen 1
Yue Guang 1
Yi Bin 2
Li Yu-jie 2 lyj09c@163.com
Shi Yuan 1 shiyuan@hospital.cqmu.edu.cn
1 Department of Neonatology, Children’s Hospital of Chongqing Medical University, National Clinical Research Center for Child Health and Disorders, Ministry of Education Key Laboratory of Child Development and Disorders, China International Science and Technology Cooperation Base of Child Development and Critical Disorders, Chongqing Key Laboratory of Childhood Nutrition and Health , Chongqing , China
2 Southwest Hospital, Third Military Medical University, Chongqing , Chongqing , China
Gould Gwyn
Electronic publication date: 2024 Jan 16
Publication date: 2024
Volume: 12
Electronic Location ID: e16744
Received 2023 Sep 13; Accepted 2023 Dec 11
Copyright: © 2024 Cao et al.
Copyright year: 2024
Copyright holder: Cao et al.
License: This is an open access article distributed under the terms of the Creative Commons Attribution License, which permits unrestricted use, distribution, reproduction and adaptation in any medium and for any purpose provided that it is properly attributed. For attribution, the original author(s), title, publication source (PeerJ) and either DOI or URL of the article must be cited.
License URL: https://creativecommons.org/licenses/by/4.0/

Keywords: Sphingosine 1-phosphate receptor 2, Cholestatic liver disease, Conjugated bile acid, Gut microbiome

Funding: National Natural Science Foundation of China 82070630, 82100658, 82270656 and 81870422 Chongqing Talents Project CQYC202103080 Chongqing PhD Project CSTB2022BSXM-JCX0006 Chongqing Natural Science Fund CSTB2022NSCQ-MSX0211 This study was supported by the National Natural Science Foundation of China (No. 82070630; No. 82100658; No. 82270656 and No. 81870422). Chongqing Talents Project (No. CQYC202103080), the Chongqing PhD Project (CSTB2022BSXM-JCX0006), and the Chongqing Natural Science Fund (CSTB2022NSCQ-MSX0211). The funders had no role in study design, data collection and analysis, decision to publish, or preparation of the manuscript.

==============================
Aims

The objective of this study is to examine the impact of inhibiting Sphingosine 1-phosphate receptor 2 (S1PR2) on liver inflammation, fibrogenesis, and changes of gut microbiome in the context of cholestasis-induced conditions.

Methods

The cholestatic liver injury model was developed by common bile duct ligation (CBDL). Sprague-Dawley rats were randomly allocated to three groups, sham operation, CBDL group and JTE-013 treated CBDL group. Biochemical and histological assessments were conducted to investigate the influence of S1PR2 on the modulation of fibrogenic factors and inflammatory infiltration. We conducted an analysis of the fecal microbiome by using 16S rRNA sequencing. Serum bile acid composition was evaluated through the utilization of liquid chromatography-mass spectrometry techniques.

Results

In the BDL rat model, the study findings revealed a significant increase in serum levels of conjugated bile acids, accompanied by an overexpression of S1PR2. Treatment with the specific inhibitor of S1PR2, known as JTE-013, resulted in a range of specific effects on the BDL rats. These effects included the improvement of liver function, reduction of liver inflammation, inhibition of hepatocyte apoptosis, and suppression of NETosis. These effects are likely mediated through the TCA/S1PR2/NOX2/NLRP3 pathway. Furthermore, the administration of JTE-013 resulted in an augmentation of the diversity of the bacterial community’s diversity, facilitating the proliferation of advantageous species while concurrently inhibiting the prevalence of detrimental bacteria.

Conclusions

The results of our study suggest that the administration of JTE-013 may have a beneficial effect in alleviating cholestatic liver disease and restoring the balance of intestinal flora.

Introduction

Cholestatic liver disease, including primary sclerosing cholangitis (PSC) and primary biliary cholangitis (PBC), has been identified as a major cause of liver transplantation (Hasegawa et al., 2021). The availability of effective treatments is limited, and the underlying molecular mechanisms remain poorly understood. The common bile duct ligation (CBDL) model has been widely recognized as a valuable tool for studying the molecular mechanisms underlying cholestatic cirrhosis (Andrade et al., 2009). In the pathogenesis of the cholestatic liver disease, the composition of bile acids plays a crucial role. Specifically, the accumulation of conjugated bile acids (CBAs) has been identified as a risk factor for the development of cirrhosis and its associated complications (Sydor et al., 2020; Wang et al., 2016; Weiss et al., 2016; McMillin et al., 2016; Stepien et al., 2021).

Sphingosine 1-phosphate receptors (S1PRs) are extensively involved in various biological processes, including the regulation of vascular and bronchial tone, vascular permeability, and the modulation of lymphocyte trafficking. Currently, the FDA has approved four S1PR modulators (fingolimod, siponimod, ozanimod, and ponesimod) for the treatment of multiple sclerosis (McGinley & Cohen, 2021). Sphingosine 1-phosphate receptor 2 (S1PR2) has been found to be involved in various biological processes, including immune cell migration (Xiong et al., 2019), maintenance of endothelial barrier integrity (Wang et al., 2022), regulation of smooth muscle cell function (Liu et al., 2020), the development and integrity of epithelial hair cell (Zhou et al., 2022). However, the clinical significance of S1PR2 functions remains less well established, and further investigation is required to elucidate its specific mechanism.

S1PR2 exhibits selective activation by CBAs, particularly taurocholic acid (TCA) while remaining unresponsive to unconjugated bile acids (Liao et al., 2023). Previous studies have provided insights into the notable role played by the S1P/S1PR2 signaling pathway in liver inflammation and fibrosis (Hou et al., 2020). For instance, the role of S1PR2 in promoting inflammation initiation and liver fibrosis development through the NLRP3 pathway has been demonstrated (Hou et al., 2021a). The activation of S1PR2 has also been associated with the stimulation of macrophages (Wang et al., 2023; Yang et al., 2023; Yang et al., 2018) and the induction of neutrophil extracellular traps (NETs) (Hirao et al., 2023; Zhao et al., 2021; Zhao et al., 2020), resulting in liver inflammation and fibrosis. Moreover, previous research has demonstrated that the absence of S1PR2 has a notable impact on the proliferation of cholangiocytes and the development of cholestatic injury caused by bile duct ligation (Wang et al., 2017a; Yang et al., 2023).

In relation to the aforementioned pathological effects, our proposal suggests that the accumulation of CBAs in the CBDL rat model facilitates the advancement of liver injury and inflammation through the activation of the S1PR2 signaling pathway. Furthermore, our objective is to provide a comprehensive understanding of the molecular mechanism through which S1PR2 modulates liver injury and inflammation.

Materials and Methods

Animals and experimental design

The protocol of this study were approved by the animal research committee from the Third Military Medical University (AMWEC20224009), and they all complied with the National Research Council’s Guide for the Care and Use of Laboratory Animals. We obtained a total of 45 male Sprague-Dawley rats (weighing 180–220 g) from the Laboratory Animal Center of the Third Military University. The rats were bred in a controlled environment that was free from specific-pathogen-free (SPF). They were provided with sterilized water and a standard diet for rats. All surgical procedures were conducted utilizing an isoflurane chamber, and anesthesia was consistently administered during the entirety of the experiment using isoflurane (R510-22; RWD, Shenzhen, China) delivered through a small animal anesthesia machine’s nose cone (R500IE; RWD, Shenzhen, China). JTE-013, a selective antagonist of S1PR2, was acquired from MedChemExpress (Cat# HY-100675; Med Chem Express, Hong Kong, China). Three groups were randomly formed by dividing the animals: The first group, referred to as the sham-operation group, consisted of rats that underwent sham surgeries without bile duct ligation (n = 15). The second group, known as CBDL group, underwent CBDL model by performing bile duct ligation (BDL) near the junction of the hepatic ducts and above the entrance of the pancreatic duct using 5–0 silk. Then the ligatures were tightened, and the common bile duct was resected between the ligatures, n = 15) and the CBDL + JTE-013 group (CBDL rats treated with intraperitoneal administration of JTE-013 at a dosage of 10 mg/kg, two injections each week for a duration of 2 weeks, n = 15). At the 3 weeks after surgery, there were 15 surviving rats in the sham group, 13 in the CBDL group, and 11 in the JTE-013 group. Two rats were excluded from the study due to failed modeling, both of these rats belonged to the JTE-013 group. Four rats were excluded from the study due to inadequate fecal collection. Specifically, three rats from the sham group and one rat from the JTE-013 group were affected. Consequently,16s rRNA sequencing was conducted on a total of 33 rats, comprising 12 rats in the sham group, 13 rats in the CBDL3w group, and eight rats in the JTE-013 group).

Sample collection

Fecal samples were collected 3 weeks after surgery in each group. Fecal samples were obtained within a SPF environment, transferred into sterile 1.5 mL container, and promptly submerged in liquid nitrogen within 30 min. The frozen fecal samples were stored at −80 °C for subsequent analysis on the gut microbiota. In the 3 weeks following CBDL, rats were anesthetized by inhalation of isoflurane and sacrificed to collect blood from the abdominal aorta. Blood was centrifuged at 3,000 g for 20 min at 4 °C to separate plasma. The resulting plasma was then divided into four aliquot: one aliquot was used for liver function testing, another aliquot was used for Enzyme-linked immunosorbent assay (ELISA) assay, a third aliquot was used for serum bile acid quantification, and the remaining aliquot was stored for future use. Rat livers were extracted and a small portion of the liver was immediately preserved in 10% neutral-buffered formalin for histological examination. The remaining liver tissue was rapidly frozen in liquid nitrogen for subsequent molecular analysis.

Histology

Hematoxylin and eosin (HE) (G1120; Servicebio, Beijing, China) and Sirius Red Staining Kit (G1340; Solarbio, Beijing, China) were used to stain the rat liver sections that had been paraffin-embedded. Immunochemical staining was conducted using an anti-CD31 antibody (ab119339; Abcam, Cambridge, UK, 1:100) to provide evidence of angiogenesis. Anti-CK-19 antibody (GB11197; Servicebio, China, 1:250) was used for assessing bile duct epithelial cell proliferation. Anti-Collagen I (GB11022-3; Servicebio, China, 1:1,000) was performed to provide evidence of fibrosis. Anti-Caspase-3 Rabbit pAb (GB11009-1; Servicebio, China, 1:200 dilution) and TUNEL staining (GDP1042; Servicebio, China) were employed to assess apoptosis. To evaluate the occurrence of NETosis, a co-staining procedure was conducted using CitH3 (ab281584; Abcam, Cambridge, UK) and MPO (ab208670; Abcam, Cambridge, UK). Additionally, the anti-S1PR2 antibody (DF4921; Affinity, San Francisco, CA, USA, 1:100) was employed. The nuclei were stained using DAPI (G1012; Servicebio, Beijing, China). For each immunofluorescence slice, ten field were randomly selected using the Pannoramic MIDI (3DHISTECH, Hungary). The quantification of positive cells per high-power field was performed using Image-Pro Plus software (version 6.0; Media Cybernetics Inc, Rockville, MD, USA). For immunohistochemistry, the liver tissue was cut into 5 μm slices, which were subsequently subjected to dewaxing using xylene and rehydrated with distilled water in an ethanol gradient. Following a 30-min incubation with 3% hydrogen peroxide (H2O2) at room temperature, the sections were subsequently blocked with 1% bovine serum albumin. They were then treated with the following primary antibodies: anti-S1PR2 (DF4921; Affinity, San Francisco, CA, USA, 1:100), CD163 (ab182422; Abcam, Cambridge, UK, 1:500), MCP-1 (ab7202; Abcam, Cambridge, UK, 1:300), IL-6 (ab208113; Abcam, Cambridge, UK, 1:500), TNF-α (ab6671; Abcam, Cambridge, UK, 1:100), NOX2 (19013-1-AP; Proteintech, Rosemont, IL, USA, 1:1000) or NLRP3 (27458-1-AP; Proteintech, Rosemont, IL, USA, 1:200) for an entire night at 4 °C. Following this, an incubation process was carried out with, a secondary antibody. Finally, following the incubation of slices with 100 L of 3,3′-diaminobenzidine reagent, a subsequent staining procedure was performed using hematoxylin.

Analysis of hepatic function

The serum of the rats was utilized to collect liver biochemistry data, including aspartate aminotransferase (AST), total bile acid (TBA), alanine transaminase (ALT), and alkaline phosphatase (ALP). This data was used as evidence of liver injury and was analyzed using the Automatic Biochemistry Analyzer (AU5400; Olympus, Tokyo, Japan).

Western blot

Total proteins were extracted from liver tissue using a total protein extraction kit for animal cultured cells and tissues (SD-001/SN-002; Invent Biotechnologies, Plymouth, MN, USA). The protein concentration was determined using the BCA method (SF247582; Thermo Fisher Scientific, Waltham, MA, USA). Western blot analysis was performed using specific primary antibodies, including α-SMA (ab7817; Abcam, Cambridge, UK), and VEGFR1 (ab184784; Abcam, Cambridge, UK), and secondary antibody GAPDH (10494-1-AP; Proteintech, Rosemont, IL, USA). The immunoreactive bands were visualized using an ECL chemiluminescent kit (XF345252; Thermo Fisher Scientific, Waltham, MA, USA). After removing background noise and adjusting for loading controls, band density was quantified using Image J software (version 1.5.3; WS Rasband, National Institute of Health, Bethesda, MD, USA).

Enzyme‑linked immunosorbent assays (ELISA)

The level of growth factors, chemotactic factor, and inflammatory cytokines in the serum were assessed using a rat ELISA kit (Shanghai Jianglai Biological Technology, Shanghai, China). The specific factors measured included placental growth factor (PLGF) (JL11559), vascular endothelial growth factor (VEGF) (JL21369), Chemokine CXCL2 (CXCL2) (JL21265), Monocyte Chemoattractant Protein 1 (CCL2) (JL11450), Tumor Necrosis Factor-alpha (TNF-α) (JL13202), Nuclear Factor-kappa B (NF-κB) (JL21039), interleukin-6 (IL-6) (JL20896). All procedures were conducted in accordance with the instructions provided by the manufacturer. The color’s intensity at 450 nm of absorbance was measured using the Thermo reader (Varioskan Flash multimode reader; Thermo Scientific, Waltham, MA, USA). The concentration of these factors was expressed in picograms per milliliter (pg/ml), except for IL-6 which was expressed in nanograms per milliliter (ng/ml).

Gut microbiota 16S rRNA sequencing

A total of 200 mg of fecal samples were used for DNA extraction according to the manufacturer’s instructions. The research conducted by Zhang et al. (2023) focuses on the investigation of the hypervariable region, as well as the primers, and laboratory techniques employed for PCR amplification and quantification. The SILVA 16S rRNA database (v138) and the Naive Bayes consensus taxonomy classifier, developed in Qiime2, were employed to accurately classify ASVs into their respective taxonomic groups. To assess the abundance and diversity of gut microbiota across the three groups, the total number of sequences for each level of taxonomic (e.g., phylum and genus) were computed and aggregated. The bioinformatic study of the gut microbiota was conducted using the Majorbio Cloud platform (https://cloud.majorbio.com). Mothur version 1.30.1 was utilized to calculate rarefaction curves and alpha diversity indices using the data obtained from the Amplicon Sequence Variants (ASVs). The computed indices included observed ASVs, Chao1 richness, Shannon index, and Simpson index, ACE, observed species. By employing Vegan v2.5-3 package’s principal coordinate analysis (PCoA) utilizing Bray-Curtis dissimilarity,a comparison of microbial communities across different samples was facilitated. The Vegan v2.5-3 package was utilized to perform the PERMANOVA test,which aimed to assess the proportion of variance attributed to the therapy and its statistical significance. The raw data were uploaded to the NCBI Sequence Read Archive (SRA) database (BioProject ID: PRJNA1012444).

Bile acids quantification in serum

An LC-ESI-MS/MS (UHPLC-Qtrap) was utilized in the present study to detect the target compounds in sample both qualitatively and quantitatively. A Waters BEH C18 (150*2.1 mm, 1.7 m) liquid chromatography column (AB SCIEX) was used to separate the analyte chemicals. The flow rate of the mobile phases, which were given at 0.35 mL/min, was 0.1% formic acid-acetonitrile solution (solvent B) and 0.1% formic acid-water solution (solvent A). Each ion fragment was automatically identified and integrated using default parameters in the AB Sciex quantitative software OS, and manual inspection was aided.

Statistical analysis

Continuous data was represented by the mean with standard deviation (SD) or the median (interquartile range) based on the distribution of the data. The Mann-Whitney U-test was employed to compare the species variation between the two groups. One-way ANOVA was employed to assess the variations among the three groups, and the least significant difference (LSD) method was utilized for the post hoc test. A significance level of 0.05 was adopted as the threshold for statistical significance in all two-sided statistical tests. All statistical analyses were performed using SPSS software (version 25.0), and all bar charts were generated using GraphPad PRISM (version 7.00; GraphPad Software, San Diego, CA, USA).

Results

Alterations in the serum bile acid profile and intestinal flora in rats with cholestatic liver injury induced by CBDL

H&E staining of liver tissue demonstrated that at 3 weeks after CBDL, significant liver injuries were prominently observed in the rats, including bile duct proliferation and dilation, lobular structure destruction, cirrhotic nodule formation, and inflammatory cell infiltration (Fig. 1A). Meanwhile, the serum bile acid profile exhibited significant alterations in the CBDL rats. As anticipated, the ligation of the bile duct prevented the primary binding bile acids (TCA, TCDCA) synthesized by the liver from entering the intestine. Consequently, the enterohepatic circulation was disrupted, leading to a significant decrease in both primary and secondary bile acids in the stool due to the obstruction of their source. This process leads to a notable elevation in the levels of primary conjugated bile acids (such as TCA, TCDCA, and T-β-MCA) in the blood. The observed alterations in the spectra of serum bile acids were found to be in accordance with our initial hypothesis, as depicted in Fig. 1B.

Figure 1 Serum bile acid profile and intestinal flora were changed after CBDL.

(A) Successful establishment of models for cholestasis-induced liver fibrosis. Representative images of H&E staining between sham group and CBDL group after 3 weeks of operation (scale bar = 50 um). (B) Serum targeted bile acids indicate a significant increase in conjugated bile acids in the CBDL group, compared to the sham group. (C) Immunohistochemistry results demonstrate a significant increase in S1PR2 expression in the CBDL group (scare bar = 20 um). (D) Semi-quantitative analysis for S1PR2 OD value. (E) Western blot results indicate a significant increase in liver S1PR2 expression in the CBDL group. (F) Quantitative analysis for S1PR2 protein. (G) The α-diversity of gut microbiota was determined using Chao, Sob, and Simpson indices (sham group; n = 12; CBDL group, n = 13). (H) Unweighted UniFrac-based principal coordinate analysis (PCoA) based on the ASV levels (sham group; n = 12; CBDL group, n = 13). (I) Microbial composition analysis at the phylum level in two groups of samples. (J) Differential microbiota between the sham group and the CBDL 3w group. CA = Cholic acid; CDCA = Chenodeoxycholic acid; TCA = Taurocholic acid; TCDCA = Taurochenodeoxycholic acid; DCA = Deoxycholic acid; LCA = Lithocholic acid; HCA = hyocholic acid. *P < 0.05, **P < 0.01, ***P < 0.001.

Interestingly, the hepatic expression of S1PR2, a receptor that specifically binds to the conjugated bile acid, exhibited a significant increase the CBDL 3w group’s liver (Figs. 1C–1F). Compared to the sham group, the CBDL 3w group exhibited a significant decrease in the biodiversity of intestinal flora. This was supported by the observed decrease in α-diversity levels (Fig. 1G). Additionally, there was a significant difference in the gut microbiota between the CBDL 3w group and the sham group (Fig. 1H). Despite the fact that the phylum Firmicutes is the most abundant in both of these groups, there was a relative increase in Proteobacteria, Actinobacteriota, and Bacteroidota after CBDL (Fig. 1I). At the genus level, the CBDL 3w group exhibited a higher proportion of Clostridium_sensu_stricto_1, Bifidobacterium, and Dubosiella, while showing a lower abundance of Bacillus, UCG-005, Romboutsia, Lachnospiraceae_NK4A136_group, Ruminococcus and Christensenellaceae_R-7_group, in comparision to the sham group (Fig. 1J).

JTE-013 ameliorates the CBDL-induced cholestatic liver injury

An investigation was conducted to determine if the specific intervention of S1PR2 could effectively suppress liver injury caused by CBDL and subsequently restore normal liver function. JTE-013, a selective antagonist of the S1PR2, was administered to rat models with CBDL, as depicted in Fig. 2A. As anticipated, the intervention of JTE-013 remarkably ameliorated cholestatic liver injury, as demonstrated by the decrease in levels of ALT (sham vs. CBDL vs. CBDL treated with JTE-013 = 30.5 ± 2.70 vs. 181.5 ± 45.73 vs. 98.5 ± 27.62 (U/L), P < 0.01), and ALP (148.8 ± 26.71 vs. 629.6 ± 256.96 vs. 356.0 ± 55.27 (U/L), P < 0.05) (Fig. 2B). Additionally, the histological injury was ameliorated (Fig. 2E); fibrosis was reduced (Figs. 2C–2F and 2H, P < 0.001), and the proliferation of bile duct was diminished (Figs. 2E and 2G, P < 0.001). Following the administration of JTE-013, a significant reduction in liver inflammation was observed, this was demonstrated by changes in ELISA measurements of NF-κB (sham vs. CBDL vs. CBDL treated with JTE-013 = 168.5 ± 26.9 vs. 1123.2 ± 323.8 vs. 676.5 ± 99.7 (pg/ml), P < 0.01), TNF-α (0.08 ± 0.01 vs. 0.14 ± 0.03 vs. 0.07 ± 0.01 (pg/ml), P < 0.01), CXCL2 (10.4 ± 2.6 vs. 48.9 ± 7.5 vs. 35.7 ± 7.4 (pg/ml), P < 0.01), CCL2 (41.7 ± 32.3 vs. 2068.1 ± 843.1 vs. 825.9 ± 437.7 (pg/ml), P < 0.01) (Fig. 3A). Additionally, immunohistochemical analysis revealed changes in CD163, MCP1, IL-6, and TNF-α (Figs. 3B and 3C, all P < 0.05). Simultaneously, there was a notable decrease in hepatocyte apoptosis, as indicated by the decrease in Tunel-positive cells (Figs. 3D and 3E, P < 0.001) and Caspase 3-labeled cells (Figs. 3D and 3F, P < 0.001). Furthermore, liver angiogenesis was also ameliorated after the injection of JTE-013, as evidenced by ELISA changes of PLGF (30.3 ± 6.9 vs. 1085.4 ± 257.0 vs. 632.1 ± 307, P < 0.01) and VEGF (34.4 ± 2.7 vs. 376.8 ± 189.4 vs. 104.3 ± 23.3, P < 0.01) (Fig. 4A), number changes of CD31-labelled blood vessel (Fig. 4B, P < 0.001), as well as protein level changes of VEGFR1 (Fig. 4C, P = 0.01). These results strongly suggest that JTE-013 attenuates cholestatic liver injury possibly by reducing the hepatic inflammatory response and apoptosis.

Figure 2 The use of JTE-013 can alleviate liver injury and fibrosis.

(A) Animal experimental procedure diagram. (B) The JTE-013 group shows alleviated liver dysfunction compared to the CBDL group. (C) Western blot results of the fibrosis marker α-SMA. (D) Quantitative analysis for α-SMA protein (E) Representative images of H&E staining (scale bar = 50 um), Masson (scale bar = 100 um), and Immunofluorescence (scale bar = 10 um) from 3 group. (F) METAVIR score for fibrosis. (G) Semi-quantitative analysis for CK-19 mean OD value. (H) Semi-quantitative analysis for Col-I mean OD value.

Figure 3 JTE-013 significantly reduces hepatic infiltration of inflammatory cells and the release of inflammatory cytokines, mitigating hepatocyte apoptosis.

(A) Serum inflammatory chemokines and cytokines. (B) Immunohistochemistry shows a significant decrease in macrophage markers (CD163, MCP1) after treatment with JTE-013 (scale bar = 100 um). (C) Immunohistochemistry shows a significant decrease in inflammatory cytokines (IL-6 and TNF-α) after treatment with JTE-013 (scale bar = 100 um). (D) Immunofluorescence results show a significant decrease in hepatocyte apoptosis after the use of JTE-013 (scale bar = 20 um). (E) Semi-quantitative analysis for TUNEL mean OD value. (F) Semi-quantitative analysis for Caspase-3 mean OD value. *P < 0.05, **P < 0.01.

Figure 4 JTE-013 can alleviate hepatic angiogenesis.

(A) Serum growth factors. (B) JTE-013 can reduce angiogenesis shown in immunohistochemical staining of CD31 as a marker for angiogenesis (scale bar = 20 um). (C) Representative images of VEGFR1 from three groups. (D) Quantitative analysis for VEGFR1 protein. ***P < 0.001, ****P < 0.0001.

JTE-013 reverses the dysbiosis of gut microbiota induced by CBDL

According to changes in α diversity, the CBDL 3w group treated with JTE-013 showed improvements in the abundance of intestinal flora compared with the CBDL 3w group (Fig. 5A). The PCoA analysis revealed significant differences between the CBDL and JTE-013 treated CBDL groups (P = 0.01, R = 0.23) (Fig. 5B). According to the community barplot analysis, the group treated with JTE-013 for CBDL showed a reduction in both Proteobacteria and Actinobacteriota (Fig. 5C). At the genus level, the JTE-013 treated CBDL 3w group exhibited a higher abundance of the genera norank_f_Muribaculaceae, NK4A214_group, Ruminococcus, and Christensenellaceae_R-7_group, while showing a lower abundance of Enterococcus compared to the CBDL 3w group (Fig. 5D). In summary, the JTE-013 treatment resulted in an increased abundance of fecal flora, a higher amount of beneficial bacteria, and while a decreased amount of harmful bacteria.

Figure 5 Effects of JTE-013 on the gut microbiota in CBDL rats.

(A) JTE-013 improve the abundance of intestinal flora CBDL group, n = 13; JTE-013 group, n = 8. (B) Unweighted UniFrac-based principal coordinate analysis (PCoA) based on the ASV levels CBDL group, n = 13; JTE-013 group, n = 8. (C) Microbial composition analysis at the phylum level between CBDL group and JTE-013 group. (D) Differential microbiota between the JTE-013 3w group and the CBDL 3w group. *P < 0.05, **P < 0.01.

Inhibition of S1PR2 by JTE-013 blocks NETosis and reduces the liver damage through the TCA/S1PR2/NOX2/NLRP3 pathway

Excessive NETosis has been verified to be associated with chronic inflammation. MPO and CitH3 have been used as markers for NETosis. Immunofluorescence results revealed a significant increase in the number of cells co-stained by MPO (Fig. 6F, P = 0.009) and CitH3 (Fig. 6G, P < 0.01) in the CBDL group, indicating an elevated occurrence of NETosis. Conversely, in the CBDL group treated with JTE-013, a significant decrease in the number of co-stained cells was observed. This indicates that JTE-013 has the ability to reduce the occurrence of NETosis and consequently inhibit inflammation (Fig. 6E). The expression levels of NOX2 (Figs. 6A and 6B, P < 0.05) and NLRP3 (Figs. 6C and 6D, P < 0.001) were increased in the CBDL group. However, both markers showed a significant reduction after JTE-013 treatment. Additionally, the treatment with JTE-013 also resulted in a decrease in the index of oxidative stress MPO protein, compared to the CBDL group (Figs. 6H and 6I, P < 0.001). These findings suggest that TCA might mitigate NETosis and liver injury through the S1PR2/NOX2/NLRP3 pathway.

Figure 6 JTE-013 can reduce the expression of NOX2 and the occurrence of NETosis.

(A) Representative images of NOX2 Immunohistochemistry staining from three groups (scale bar = 50 um). (B) Quantitative analysis for NOX2 positive area%. (C) Representative images of NLRP3 Immunohistochemistry staining from three groups (scale bar = 50 um). (D) Quantitative analysis for NLRP3 positive area%. (E) Immunofluorescent double staining of MPO and CitH3 in liver tissue (scale bar = 10 um). (F) Quantitative analysis for CitH3 mean OD value. (G) Quantitative analysis for MPO mean OD value. (H) Western blot results of MPO. (I) Quantitative analysis for MPO protein. *P < 0.05, **P < 0.01, ***P < 0.001, ****P < 0.0001.

Discussion

Our study revealed a significant increase in both conjugated bile acids and S1PR2 level following CBDL. Inhibiting S1PR2 by its antagonist JTE-013 resulted in several beneficial effects. These effects included the restoration of intestinal flora balance, improvement in liver function, reduction of liver inflammation and hepatocyte apoptosis, and suppression of NETosis. These effects were possibly mediated through the TCA/S1PR2/NOX2/NLRP3 pathway.

In this study, we used JTE-013 as a selective antagonist of S1PR2. However, a recent study conducted by Pitman et al. (2022) provided that JTE-013 could bind to S1PR4 and may have potential off-target effects. Therefore, we further explored the specificity of JTE-013. We found that S1PR4 is less expressed in the liver (Fig. S1B), S1PR4 was not inhibited after JTE-013 administration according to the results of qPCR (Fig. S1C), and S1PR2 Si-RNA markedly inhibited pro-inflammation activation by TCA in BRL-3A, which is consistence with that in JTE-013 treated CBDL rats (Fig. S1D). Although we have done verification, it is necessary to consider non-specific or off-target problems when using inhibitors.

Numerous organs have been associated with sphingosine-1-phosphate (S1P) and its receptor S1PR2, which elicit a wide range of effects on physiological and pathological processes. The S1P/S1PR2 signaling axis plays a crucial role in refulating cardiovascular system, encompassing the modulation of vascular tone, maintenance of endothelial barrier integrity, and promotion of angiogenesis (Pan et al., 2021). In the context of the immune system, the S1P/S1PR2 signaling axis was found to have significant effects on immune cell trafficking, lymphocyte egress from lymphoid organs, as well as the development and function of immune cells (Cartier & Hla, 2019; Muppidi et al., 2014). In the context to the pulmonary system, the signaling pathway involving S1P/S1PR2 has been linked to various physiological processes such as airway smooth muscle contraction, inflammation, and lung fibrosis (Bahlas et al., 2020; Choi, Kim & Kinet, 1996). Furthermore, the S1P/S1PR2 signaling axis has been identified as playing a role in various hepatic processes, including liver injury, fibrosis, and cholestasis (Hou et al., 2020; Alqarni et al., 2019; Wang et al., 2017b). Collectively, the aforementioned findings indicate that S1PR2 holds promise as a potential therapeutic target for a range of diseases. However, the activation of S1PR2 is not limited to S1P alone, as it can also be specifically activate through its binding with conjugated bile acids (Hylemon et al., 2021). The signaling pathway involving TCA/S1PR2 has received limited research attention. In our study, it was observed that the administration of JTE-013 resulted in a significant decrease in proliferation of cholangiocytes. Our finding are in line with previous studies conducted by Wang et al. (2017a) and Liu et al. (2014). Furthermore, the JTE-013 treatment exhibited hepatoprotective effects by mitigating liver injury and fibrosis in cholestatic liver cirrhosis, corroborating the conclusion from Liao et al. (2023).

The mechanisms underlying this phenomenon are complex and multifaceted. According to previous literature, the activation of S1PR2 was found to promote inflammation and liver fibrosis through the NLRP3 pathway, as well as modulate neutrophil exocytic reticulum (Hou et al., 2021a; Hou et al., 2020). Excessive production of neutrophil extracellular traps (NETs) has been linked to the development of chronic inflammation (Papayannopoulos, 2018). Therefore, we conducted an examination of indicators related to NLRP3 and NETosis in order to investigate their potential pathways. In summary, our study aims to investigate, for the first time, the role of the TCA/S1PR2 pathway in NETosis. In this study, in order to further investigate whether TCA or S1P activates the action of S1PR2, we explored the level of S1P in the liver. As predicted, S1P in the liver increased in response to inflammation, but unfortunately, it was not statistically significant (Fig. S2A), and the correlations between the liver level of S1P and the liver function were not significant (Fig. S2B). In summary, TCA was more important for activating S1PR2 in cholestatic liver injury. Our study observed that a significant increase in neutrophil infiltration and a heightened occurrence of NETosis during CBDL. The JTE-013 treatment, however, demonstrated a reduction in neutrophil infiltration and the development of NETosis. Consistent with our findings, previous studies have presented compelling evidence that supports the remarkable reduction in neutrophil infiltration during the BDL-induced liver injury following the blockade of the S1PR2 receptor (Zhao et al., 2021). Secondly, it is crucial to note that NADPH oxidase enzymes (NOX) play a critical role in the ROS pathway, specifically in the process of NETosis (Papayannopoulos, 2018). Several studies have already explored the relationship between NOX2 and NETosis. For example, Leung et al. (2021) provided that the administration of NOX2 inhibitors in the treatment of heparin-induced thrombocytopenia can effectively prevent NETosis and reduce the occurrence of thrombosis. The study from Li et al. (2022) demonstrated that colchicine effectively inhibited the formation of neutrophil extracellular traps (NETs) by reducing the production of NOX2/ROS and the influx of Ca2+. In our study, we observed an increase in the expression levels of NOX2 following CBDL, whereas a significant reduction was observed after treatment with JTE-013. The administration of JTE-013 resulted in a decrease in MPO levels, which is indicative of a reduction in oxidative stress. The findings of this study indicate that the activation of S1PR2 may play a role in promoting NETosis, potentially through the involvement of NOX2. Thirdly, according to the study of Al-Kuraishy et al. (2023) elevated levels of macrophage S1PR2 expression result in the release of pro-inflammatory cytokines and the activation of NLRP3 inflammation. According to Hou et al. (2020) blocking the macrophage S1PR2 receptor reduced the NLRP3 inflammasome-induced hepatic fibrogenesis and inflammation. Many studies have demonstrated that inhibiting the NOX2/NLRP3 signaling axis can reduce inflammatory damage (Zhang et al., 2022) and liver fibrosis (Wan et al., 2022). In summary, the findings of our study demonstrate that the administration of JTE-013 has the potential to decrease the activation of NOX2 and NLRP3, inhibit NETosis formation, and suppress the production of pro-inflammatory cytokines (such as NF-κB, TNF-α, and IL-6). These results align with the conclusions reported by Hylemon et al. (2021). Therefore, we hypothesized whether the S1PR2 facilitates the process of NETosis and contributes to liver damage through the TCA/S1PR2/NOX2/NLRP3 signaling pathway. However, the present findings may not provide sufficient evidence to establish that S1PR2 inhibits the inflammatory response through the NOX2/NLRP3 pathway. Additional experiments involving knockout models and the assessment of additional inflammatory factors are necessary to support this hypothesis.

Growing data indicates that aberrant angiogenesis and hepatic fibrosis are related processes that co-occur (Thabut et al., 2011). Furthermore, in cirrhotic rats, anti-angiogenic treatment has effectively reduced fibrosis and portal pressure (Tugues et al., 2007). These results led us to hypothesize that antifibrotic medications might have an inhibitory effect on angiogenesis. According to our initial hypothesis, our current study confirmed that treatment with JTE-013 could slow down liver angiogenesis and reduce liver damage and fibrosis. However, most studies have reported that S1PR2 inhibits angiogenesis (Tang et al., 2020; Zhang et al., 2021; Zhou et al., 2022; Xiao et al., 2022; Lory et al., 2023), while S1PR1 and S1PR3 promote angiogenesis. Nevertheless, there are a few reports of a pro-angiogenic role for S1PR2. For instance, Mendelson et al. (2013) suggested a cooperative effect of S1PR1 and S1PR2 on regulating vascular development in zebrafish. Furthermore, in a murine myocardial infarction (MI), S1P can enhance its efficacy in stimulating angiogenesis through the S1PR2/ERK1/2-MMP-9 pathway (Chen et al., 2018). The reduction of scarring is a benefit of SphK1 overexpression, and it is believed that S1PR2 signaling plays a significant role (Aoki et al., 2019). We attribute these discrepancies to the distinct characteristics of S1PR2 in different tissues and diseases. However, there is still a limited number of investigations specifically targeting liver angiogenesis. Therefore, further research is needed to corroborate the ability of JTE-013 to reduce angiogenesis in cholestatic cirrhosis.

The gut and liver communicate through the portal vein, the biliary system, and the systemic circulation. The gut flora plays a crucial role in maintaining the homeostasis of the gut-liver axis. Intestinal translocated bacteria and their metabolites recognize immune receptors in liver cells during blood circulation, activating an inflammatory cascade, thereby leading to liver cell damage and fibrosis (Blesl & Stadlbauer, 2021). In turn, bile acids produced by the liver also regulate the composition of gut flora through their receptors (Wahlström, 2019). Unfortunately, few studies have explored the changes in intestinal flora in rats with cholestasis. In the current study, we found that serum taurocholic acid (TCA) in common bile duct ligation (CBDL) rats was significantly increased and closely positively correlated with liver injury via targeted bile acid metabolism assay (Fig. S1A). We also found that the biodiversity of the microbiota decreased in CBDL rats (Fig. 1G), and the abundance of Clostridiaceae, Bifidobacteriaceae, Enterococcaceae increased significantly, Peptostreptococcaceae, unclassified_c_Bacilli, Erysipelatoclostridiaceae, Ruminococcaceae decreased significantly at family level (Fig. S3A). To investigate the relationship between gut microbiota and serum bile salts, we performed a Spearman correlation analysis between gut microbiota and bile acids at family levels. Interestingly, the abundance of the Clostridiaceae, Bifidobacteriaceae, and Enterococcaceae was positively correlated with the serum level of TCA. Meanwhile, the abundance of the Peptostreptococcaceae, unclassified_c_Bacilli, and Erysipelatoclostridiaceae was negatively correlated with serum level of TCA (Fig. S3B). Moreover, after the treatment of S1PR2 inhibitor (JTE-013), the biodiversity of the microbiota (Fig. 5A), and the relatively most associated microbiota recovered (Fig. S3C). Then we investigated the impact of JTE-013 intervention on the intestinal microbiota composition in rats following CBDL at genus level. SCFA-producing bacteria (such as Romboutsia, Lachnospiraceae_NK4A136_group, Ruminococcus, and Christensenellaceae_R-7_group) (Li et al., 2022) are enriched in rats after JTE-013 treatment. This suggest that JTE-013 treatment may enhance the intestine’s ability to produce SCFA. According to the study of Sun et al. (2022), norank_f_Muribaculaceae have a negative correlation with numerous conjugated BAs such as TCA and TCDCA, indicating the increase in the norank_f_Muribaculaceae could decrease the conjugated BAs in the colon. Our results demonstrated that the CBDL model has an increased prevalence of potentially hazardous bacteria such as Enterococcus. Throughout the range of liver fibrosis severity, Enterococcus is noticeably overrepresented and strongly linked with serum ALT, AST, and ALP levels, according to Spearman correlation tests (Xiang et al., 2022). In summary, the strong correlation between increased serum TCA and intestinal microbiota indicated that TCA could affect intestinal flora. The dis-balanced intestinal microbiota after CBDL could recover after using JTE-013, indicating that S1PR2 is involved in microbiota regulation. To investigate the causality between the S1PR2 and the microbiota, knocking out the S1PR2 in rats should be further conducted.

Conclusion

Our findings demonstrate that conjugated bile acids and S1PR2 were significantly elevated after CBDL. When specifically blocking S1PR2, we found the improvement of the intestinal flora imbalance, alleviation of liver function, reduction of liver inflammation and hepatocyte apoptosis, and suppression of NETosis. These effects may be mediated through the TCA/S1PR2/NOX2/NLRP3 pathway.

Supplemental Information

Supplemental Information 1 Raw data.

Click here for additional data file.

Supplemental Information 2 JTE-013 can effectively inhibit S1PR2.

(A) The Spearman Correlation heatmap analysis between the liver function and taurocholic acid (TCA). (B) Immunofluorescence results show the level of S1PR4 in the liver was very low alleviate (scale bar = 20 um). (C) qPCR results showed that treatment of JTE-013 significantly decreased the level of S1PR2, however did not alter the level of S1PR4. (D) S1PR2 Si-RNA markedly inhibited pro-inflammation activation by TCA in BRL-3A (Rat Normal Hepatocytes Cells).

Click here for additional data file.

Supplemental Information 3 The expression of S1P in the liver and correlation with liver function.

(A) The level of S1P in liver tissues was determined by its ELISA assay. (B) The Spearman Correlation heatmap analysis between the liver function and liver S1P.

Click here for additional data file.

Supplemental Information 4 The correlation between bile acids and gut microbiota, S1PR2 may be involved in microbiota regulation.

(A) Differential microbiota between the sham group and the CBDL 3w group at family level. (B) The Spearman Correlation heatmap analysis between the liver function and gut microbiota at family level. (C) Differential microbiota among the sham group, the CBDL 3w group and the JTE-013 3w group at family level.

Click here for additional data file.

Supplemental Information 5 The Arrive Guidelines 2.0.

Click here for additional data file.

Supplemental Information 6 Uncropped WB image in Figure 1.

Click here for additional data file.

Supplemental Information 7 Uncropped WB image in Figure 2.

Click here for additional data file.

Supplemental Information 8 Uncropped WB image in Figure 6.

Click here for additional data file.

Supplemental Information 9 Uncropped WB image in Figure 4.

Click here for additional data file.

Abbreviation List

ALT Alanine transaminase

ALP Alkaline phosphatase

AST Alpha smooth muscle actin

BDL Bile duct ligation

CA Cholic acid

CBAs Conjugated bile acids

CCL2 Monocyte Chemoattractant Protein 1

CD31 Platelet endothelial cell adhesion molecule

CDCA Chenodeoxycholic acid

CitH3 Citrullinates histone H3

CK-19 Cytokeratin-19

Col-I Collagen I

CXCL2 Chemokine CXCL2

DAPI 4′,6-diamidino-2-phenylindole

DCA Deoxycholic acid

ELISA Enzyme-linked immunosorbent assay

HE Hematoxylin and eosin

IL-6 Interleukin-6

LCA Lithocholic acid

MCP-1 Monocyte chemoattractant protein-1

MPO Myeloperoxidase

NETs Neutrophil extracellular traps

NF-κB Nuclear Factor-kappa B

NLRP3 NOD-like receptor thermal protein domain associated protein

NOX2 Nicotinamide adenine dinucleotide phosphate oxidase 2

PBC Primary biliary cholangitis

PLGF Placental growth factor

PSC Primary sclerosing cholangitis

ROS Reactive oxygen species

S1PR2 Sphingosine 1-phosphate receptor 2

S1PR1 Sphingosine 1-phosphate receptor

S1P Sphingosine-1-phosphate

SPF Specific-pathogen-free

SD Standard deviation

α-SMA Alpha smooth muscle actin

TCDCA Taurochenodeoxycholic acid

TBA Total bile acid

TCA Taurocholic acid

TNF-α Tumor necrosis factor-α

TUNEL Terminal deoxynucleotidyl transferase (TdT) dUTP Nick-End Labeling

VEGF Vascular endothelial growth factor

VEGFR1 Vascular endothelial growth factor receptor 1

Additional Information and Declarations

Competing Interests

Author Contributions

Animal Ethics

DNA Deposition

Data Availability

The authors declare that they have no competing interests.

Huiling Cao conceived and designed the experiments, performed the experiments, analyzed the data, prepared figures and/or tables, authored or reviewed drafts of the article, and approved the final draft.

Lin Chen conceived and designed the experiments, performed the experiments, analyzed the data, prepared figures and/or tables, authored or reviewed drafts of the article, and approved the final draft.

Ziyang Zeng conceived and designed the experiments, performed the experiments, prepared figures and/or tables, and approved the final draft.

Xianfeng Wu conceived and designed the experiments, performed the experiments, prepared figures and/or tables, and approved the final draft.

Yuhao Lei conceived and designed the experiments, performed the experiments, prepared figures and/or tables, and approved the final draft.

Wen Jia conceived and designed the experiments, analyzed the data, authored or reviewed drafts of the article, and approved the final draft.

Guang Yue conceived and designed the experiments, analyzed the data, authored or reviewed drafts of the article, and approved the final draft.

Bin Yi conceived and designed the experiments, analyzed the data, prepared figures and/or tables, authored or reviewed drafts of the article, and approved the final draft.

Yu-jie Li conceived and designed the experiments, performed the experiments, analyzed the data, prepared figures and/or tables, authored or reviewed drafts of the article, and approved the final draft.

Yuan Shi conceived and designed the experiments, performed the experiments, analyzed the data, prepared figures and/or tables, authored or reviewed drafts of the article, and approved the final draft.

The following information was supplied relating to ethical approvals (i.e., approving body and any reference numbers):

The Third Military Medical University’s animal research committee (AMWEC20224009).

The following information was supplied regarding the deposition of DNA sequences:

The raw data are available at NCBI Sequence Read Archive (SRA) and figshare: PRJNA1012444.

Cao, mogo (2023). Changes of intestinal flora in cholestatic liver disease and the changes after JTE-013 intervention. figshare. Dataset. https://doi.org/10.6084/m9.figshare.24042966.v1.

The following information was supplied regarding data availability:

The raw data of the serum bile acid are available in the Supplemental File.

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
