# Peer review of "Reversal of cholestatic liver disease by the inhibition of sphingosine 1-phosphate receptor 2 signaling"

_PeerJ, doi:10.7717/peerj.16744_

## Round 0.1 · original submission · Major Revisions

I am returning this to you with two specific requests for further work and as detailed by the reviewers, a list of other issues which much be addressed.

Reviewer-2 has raised issues regarding the link to bile salts and the microbiota. More evidence is needed to support your assertion that SIPR2 is involved. As noted, a comparison between control and JTE treated rats (not ligated) is required (i.e., further analysis) to cement this point.

I am also concerned regarding the specificity of JET-013. Either a distinct inhibitor, or at the very least a structural analogue, is needed (i.e., further experimental work) to address this point..

Discussion of alternative possible agonists is required.

Please address all points in the reviewers' comments and clearly modify your paper to reflect these concerns. Note that further experimental work is required and is deemed essential.

Reviewer 1 ·

Basic reporting

In this manuscript, Dr. Guo and coauthors reported their efforts on investing the S1PR2 inhibition impact on on the cholestasis-induced liver inflammation, cholestasis-induced liver inflammation, fibrogenesis, and gut microbiome by using JTE-0123, a well known S1PR2 inhibitor. Based on their biochemical and histological assessments, they found the serum level of the conjugated bile acids was increased significantly and S1PR2 was over expressed in bile duct ligation (BDL) rat model compared with controls. When treated with JTE-013, S1PR2 inhibitor, they found that JTE-013 was able to alleviate the liver function, reduce liver inflammation, hepatocyte apoptosis, suppression of NETosis by the TCA/S1PR2/NOX2/NLRP3 pathway. They also reported that JTE-013 augmented the diversity of the bacterial community, promoting the growth of beneficial species wheras suppressing the presence of harmful bacteria. Together, they conclude that JTE-013 is able to relieve cholestatic liver disease and regulate the intestinal fora of imbalance. This work provides additional information to elucidate the function of S1PR2, it will be helpful to explore the S1PR2 modulate function for treating liver injury and inflammation and other diseases. the manuscript was organized and written well. The experiment was detailed sufficiently. After the authors make minor revision, it should be ready to be accepted to be published in this journal.
Special comments:
1. JET-013 is not a S1PR2 specific inhibitor, it also binding to S1PR4. If the authors provide addition description about the JTE-013, it may help address the JET-013 off-targeting function (Sci Rep. 2022 Jan 10;12(1):454. doi: 10.1038/s41598-021-04009-w).
2. If the authors are able to provide information or discussion that the development of S1PR2 therapeutics was far behind the S1PR2 therapeutics development and implementation, it will help the readers to understand the importance of the S1PR2 target.
3. The abbreviation list should be listed by alphabetical order.

Experimental design

the experimental desgin is clear

Validity of the findings

The data was provided clear and the significances were described well.

Additional comments

1. JET-013 is not a S1PR2 specific inhibitor, it also binding to S1PR4. If the authors provide addition description about the JTE-013, it may help address the JET-013 off-targeting function (Sci Rep. 2022 Jan 10;12(1):454. doi: 10.1038/s41598-021-04009-w).
2. If the authors are able to provide information or discussion that the development of S1PR2 therapeutics was far behind the S1PR2 therapeutics development and implementation, it will help the readers to understand the importance of the S1PR2 target.
3. The abbreviation list should be listed by alphabetical order.

·

Basic reporting

The manuscript is well written. The introduction adequately presents the subject of study and, in general the literature is correct.
Figures are relevant, their quality is enough for reviewing except for Fig. 1 which has too much information and texts, particularly in microbiota studies are difficult to read.

Experimental design

The research is within Scope of the journal. The aim is well defined and meaningful. The methods are adequate and sufficiently described. The treatment of animals has been approved by a local committee (Third Military Medical University’s animal research committee).

Validity of the findings

The authors hypothesize that the increase in primary bile salts taurocholate and taurochenodeoxycholate is responsible of many of the alteration induced by ligation through the activation of S1PR2. This hypothesis is sensible since primary bile salts are increased and they are agonist of the receptor. However, since receptor expression is increased, there are other possible agonists that could also be responsible; one of them is sphingosine 1-phosphate itself that can also be increased in an inflammatory condition. This should be discussed.
The role of bile salts is particularly doubtful in microbiota modification. Bile duct ligation impedes the arrival of bile salts to the duodenum so it is difficult to understand how could they affect microbiota via S1PR2. Obviously, it is demonstrated that bile salts alter the profile of microbiota (Gut Microbes. 2013 Sep 1; 4(5): 382–387.) but in this case the results show differences among ligated rats (no bile salt) depending on the presence of JTE-013. Probably this effect is mediated by S1P (Wollny et al Int J Mol Sci. 2017 Apr; 18(4): 741). More evidences are needed to support a role of bile salts via S1PR2 in the alteration of microbiota. Even if they do not participate, the role of S1PR2 will still be relevant and probably it would be better support by measurement of S1P. The comparison between control and JTE treated rats (not ligated) would be helpful to understand the results.
Bile duct ligation does not seem to be completely adequate as a model to study the effect of cirrhosis in microbiota. In cirrhotic patients, bile salts reach the duodenum (half of what happens in normal people) (Gastroenterology 1971 60, 491-498) whereas in bile duct ligation this do not happen.

Additional comments

The manuscript of Cao et al. presents relevant evidences about the role of sphingosine 1-phosphate receptor 2 in a rat model of bile duct ligation. The role of this receptor is sustained by the fact that its expression is significantly increased and that administration of an inhibitor ameliorates many parameters that are altered by bile duct ligation.
Minor comments
Primary bile salts increase and secondary bile salts decrease because enterohepatic recirculation is interrupted. This is something expected and a logical consequence of the model. It should be expressed in the text.
Why the administration of JTE-013 was performed twice a week. Probably the effect of the inhibitor lasts less than 3 days.
Line 166 It seems to be an error with the version Image J software. The present version is 1.53. Version 1.2 stated in the text is from December 2000.
Line 178 a passive participle is missing between “fecal samples were” and “for DNA”

---

## Round 0.2 · Minor Revisions

While I am satisfied with your written response, I would like you to incorporate elements of this discussion, particularly my points 1 and 3, into the revised manuscript more explicitly please. Once you have done this, I see no other issues.

·

Basic reporting

The authors have performed a thorough revision of the manuscript and have improved the writing

Experimental design

The authors have performed additional experiments including S1P measurement. This allow to discard and increase of this agonist as a responsible of the activation of S1PR2.

Validity of the findings

The discussion of microbiota modification induced by S1P2R have been moderated and it is less conclusive regarding the role of bile salts.

Additional comments

The authors have written a better manuscript and in the opinion of this reviewer it deserves publication.

---

## Round 0.3 · accepted · Accept

Thank you for attending to these minor remaining issues. I am happy to accept now.